# Statistical Modelling of the Annual Rainfall Pattern in Guanacaste, Costa Rica

**Rachel MacKay Altman** [1] **, Ofir Harari** [1,†]**, Nadya Moisseeva** [2] **and Douw Steyn** [2,*]

1   Department of Statistics and Actuarial Science, Simon Fraser University, Burnaby, BC V5A 1S6, Canada
2   Department of Earth, Ocean and Atmospheric Sciences, The University of British Columbia,
    Vancouver, BC V6T 1Z4, Canada
*   Correspondence: dsteyn@eoas.ubc.ca; Tel.: +1-604-364-1266
†   Current address: Cytel, Vancouver, BC V5Z 1J5, Canada.

**Abstract:** Rainfall in Guanacaste, Costa Rica, has marked wet/dry phases: the rainy season is punctuated by a short midsummer drought, and the dry season frequently has months of no rain. In this region, spring and summer rainfall peaks are important for local rain-fed agriculture and annual total for groundwater recharge and hydroelectricity production. We propose a novel model of rainfall in this region, the double-Gaussian model, which uses monthly total rainfall data collected from 1980 to 2020 from two meteorological observation stations. Our model provides an intuitive way of describing the seasonality of rainfall, the inter-annual variability of the cycle, and variability due to the monthly Oceanic Niño Index, ONI. We also consider two alternative models, a regression model with ARMA errors and a Tweedie model, as a means of assessing the robustness of our conclusions to violations of the assumptions of the double-Gaussian model. We found that the data provide strong evidence of an increase/decrease in rainfall in both temporal maxima during La Niña/El Niño (negative/positive ONI) conditions but no evidence of a decade-scale trend after accounting for ONI effects. Finally, we investigated the problem of forecasting future rainfall based on our three models. We found that when ONI is incorporated as a predictor variable, our models can produce substantial gains in prediction accuracy of spring, summer, and annual totals over naive methods based on monthly sample means or medians.

**Keywords:** precipitation; Costa Rica; statistical modelling; Oceanic Niño Index; time trends

## 1. Introduction

The Pacific coast of Central America (including the Guanacaste province of Costa Rica) is an interesting example of a region experiencing a tropical wet–dry climate. The Nicoya peninsula of Guanacaste (see Figure 1) experiences an extremely dry season from roughly September to May, and the rainy season is punctuated by a mild mid-summer drought (MSD) locally called *La Canícula* or *Il Veranillo di San Juan*. The Pacific coast of Central America is the only tropical region away from the equator that experiences this double maximum of rainfall [1]. The marked annual cycle of rainfall has profound influences on many aspects of the environment and human life, including agriculture, electricity generation, and tourism. The *FuturAgua* research project [2] was directed towards an understanding of the socio-hydrologic consequences of water availability, use, and governance in the Nicoya peninsula.

The overall intent of this report is to provide a statistically defensible model of rainfall variability in the Nicoya peninsula as a basis for applied work on water balances, use, and management and for related decision making. As rainfall variability is the primary driver of the water balance, a detailed characterization of the temporal variability of rainfall on a variety of time scales over the peninsula is fundamental to the understanding and management of water resources in the Nicoya peninsula [2]. Potential long-term (multi-decadal) rainfall trends driven by climate change will be important to water managers in

the region, and inter-annual variability of wet and dry seasons, and the MSD, must be well understood, as much of water management is driven by variability on a seasonal scale. Short-term rainfall variability driving floods or droughts is important too but will not be addressed in this work due to the unavailability of suitable data. Likewise, we do not address the spatial variability of rainfall in the study region.

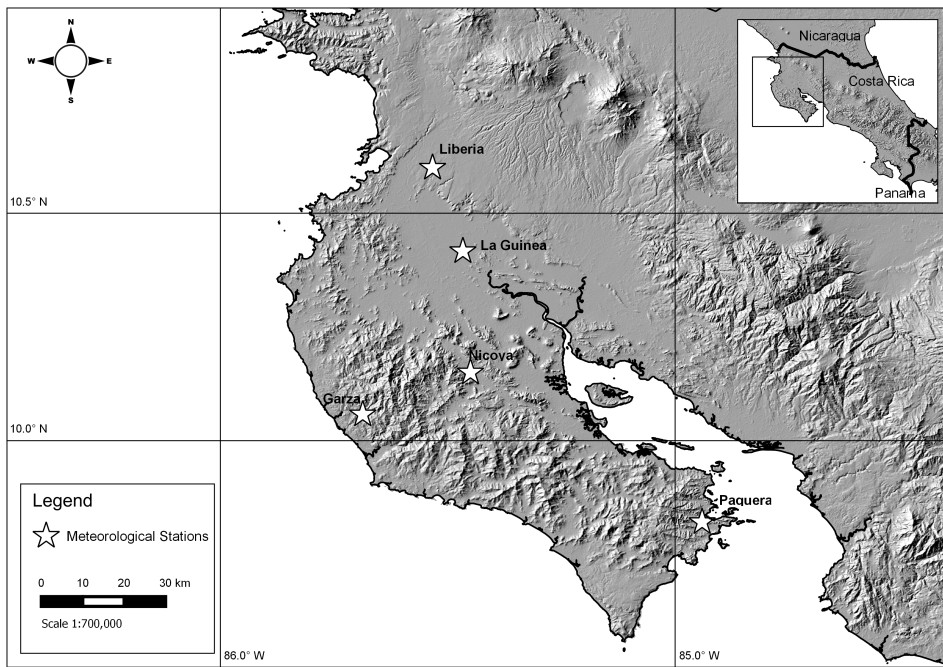

**Figure 1.** Location map of the *Futuragua* study domain showing topography and the five meteorological stations.

The underlying physical mechanisms governing the annual wet/dry rainfall pattern and the embedded MSD are not directly relevant to this work. However, a general understanding of those mechanisms will serve the work by placing the annual pattern in a climatological context and so will provide some insight into possible changes in rainfall accompanying climate change.

Previous statistical climatological studies of mechanisms, forcing, and variability underlying the annual cycle of rainfall in the wet–dry tropics of Central America have been undertaken using station data, satellite data, reanalysis data, and model output. Several authors [3,4] analyzed rainfall in Costa Rica and confirmed earlier results [5] that the annual total rainfall is related to El Niño-Southern Oscillation (ENSO); dry years generally correspond to low Southern Oscillation Index (SOI) values, also called warm phases or El Niño. A similar study [6] concluded that, in addition to ENSO effects, the relative signs of sea surface temperature (SST) anomalies in the Eastern Pacific and tropical North Atlantic Oceans have an effect on rainfall in Costa Rica. The authors noted the low levels of explained variance in their statistical analyses and poorly understood aspects of the underlying mechanisms.

The influence of large-scale atmosphere-ocean forcing by the Pacific Decadal Oscillation (PDO) and Atlantic Meriodonal Overturning (AMO) indices and North Atlantic SST on biennial variability of total rainfall in Central America has been established [7,8], but these works give no indication of such influence on the annual cycle. Therefore, although the possibility of such an influence exists, we do not address it in our study.

Some authors [9] noted a link between Central America rainfall and cross-equatorial flow and trade winds. Others [10] pointed out that this region abuts the east Pacific warm pool (EPWP) and is the rainiest place on earth in the Boreal summer. Their analysis shows statistical links among mature phase El Niño conditions, EPWP warm anomalies, enhanced

eastern Pacific ITCZ, and consequently, enhanced rainfall over Central America. These analyses show that multiple mechanisms govern rainfall in the wet/dry tropics, and not surprisingly, that statistical associations are weak.

The MSD, a particular, local feature of the annual cycle of rainfall in the wet–dry tropics of Central America, has been studied in parallel with the cycle itself. Some researchers [1] reject the idea that the MSD is associated with the double crossing of the intertropical convergence zone (ITCZ) over Central America. In agreement with earlier work [4], they postulate that the MSD is driven by an intensification of the northeast trade winds in July and August with subsequent subsidence downwind of the spine of Central America, resulting in a suppression of rainfall. The noted trade wind intensification is related to fluctuations in strength and position of the eastern Pacific ITCZ. Additional studies [11] elaborated on a mechanism relating the MSD to SST contrasts between Pacific and tropical North Atlantic Oceans—that is, a warmer Atlantic leading to less severe MSD. Another author [12] employed a satellite-derived rainfall dataset and reanalysis of winds to study the diurnal cycle of rainfall in the MSD region of Central America. His analysis shows that the diurnal cycle of rainfall during the MSD involved less evening rainfall compared to that in non-MSD months. The local circulations driving this diurnal cycle are shown to be part of the trade-wind system, thereby linking the MSD to seasonal variability of the trades. Some researchers [13] investigated possible responses of the MSD to climate change using model output from the Coupled Model Intercomparison Project Phase 3. Their analysis suggests a future early onset and increased intensity of the MSD. They emphasize that the mechanisms underlying the MSD are yet to be fully understood. Other authors [14] have examined a combination of local and remote forcings driving the MSD using satellite observations, reanalysis data, and a linear baroclinic model. They concluded that the May–June early peak is part of the southern branch of the North American Monsoon. The MSD is then driven by northward movement of the ITCZ and an associated westward extension of the Atlantic subtropical high, which drives divergence and subsidence over Central America. The August–September rainfall peak is coincident with the northward limit of the Eastern Pacific ITCZ and a peak in the Atlantic and Caribbean SST. A different group [10] suggested a simple mechanism for MSD based on the biannual crossing of solar declination in Central America. They posit that two peaks in convective instability produce the two rainfall peaks and thus reject the idea of a rainfall suppression mechanism in favor of two instances of a single precipitation enhancing mechanism. They acknowledge the influence of the variety of remote processes invoked in earlier analyses. Most recently, researchers [15] have quantified the inter-annual variability of the MSD in the entire Central American Region and shown that the MSD intensity and magnitude have a negative relationship with Niño 3.4 and a positive relationship with the Caribbean low-level jet. They have also shown that the MSD is dependent on sea surface temperature anomalies in the Pacific, Tropical North Atlantic, and Caribbean waters.

These studies show that the MSD is a persistent climatological feature of the southwest coast of Central America that is well simulated by a variety of global climate models. It is probably governed by a complex interplay of local and remote processes, including ENSO, ITCZ migration, the North American Monsoon, the North Atlantic subtropical high, tropical low-level jets, regional SST, local convection, and topographic forcing. Not surprisingly, statistical climatological analyses have failed to uncover a simple dominant governing mechanism.

Building on this previous work, our first objective was to develop an interpretable, statistically justified model of rainfall for two stations in the Guanacaste Peninsula of Costa Rica. The model had to not only capture the annual cycle but also incorporate the effects of the Oceanic Niño Index (ONI) and time (to allow for a possible climate change signal). Our second objective was to use this model as a basis for predicting rainfall in the region over a time span of one month to one year. We used available station rainfall data to inform our choice of model and to assess its predictive ability. Our approach is an extension of that used in the *FuturAgua* project, where a preliminary version of our model gave rainfall

predictions that provided essential input for the hydrological models and informed public stakeholders' engagement efforts.

Rainfall modelling is generally done in one of three distinct ways:

1. *Stochastic rainfall simulation*. These models are used to provide a statistical representation of rainfall for forecasting floods and similar events. A particular variant is used to generate synthetic sequences of rainfall for scenario building. The functional part of these models uses atmospheric thermodynamics and semi-empirical models for rainfall in various weather situations. The models are not location or time-specific and are used in both regional and point applications. Examples of such models appear in the literature [16,17].

2. *Operational rainfall modelling*. With this approach, rainfall is generated inside a fully constituted weather forecast modelling system that is run continually by many national weather services. The models are based on the fundamental constitutive equations of the atmosphere. Precipitation is modelled by semi-empirical schemes that use local thermodynamic, topographic, and weather conditions. The output, a sequence of predictions, is generally on a grid resolution of a few tens of kilometres and is produced approximately once per hour. This kind of modelling is enormously computationally intensive.

3. *General circulation modelling*. These models are also based on the fundamental constitutive equations of the atmosphere. However, in contrast with operational rainfall modelling, these models are averaged in space and time over larger areas and longer times. This type of modelling is designed to study climate change on a scale of 100 years or longer. The spatial resolution is quite coarse—tens to hundreds of kilometres. Statistical downscaling approaches can be used to produce finer-resolution (regional scale) rainfall predictions. Similar to operational modelling, this kind of modelling is enormously computationally intensive. The literature [18] provides a description of this approach to precipitation modelling.

The modelling approach we take in this work is neither stochastic nor operational. Instead, we developed a simple but novel analytical representation of rainfall seasonality from time series of observations at two stations in the Nicoya peninsula. The statistical (empirical) model that we propose describes the annual cycle of rainfall as a function of time and ENSO via ONI, a large-scale climate driver, namely, an index measure of a particularly important (in this region) component of intermediate-term climate variability. Our model is applicable only in a particular region (the wet–dry climate region of Central America), has the ability to predict, and can be used in a scenario-building mode.

The remainder of the paper is organized as follows. In Section 2, we describe the rainfall data and illustrate some of their key features. In Section 3, we propose the double-Gaussian model to describe the rainfall data. We interpret and estimate its parameters, including the effects of ONI on rainfall and decade-scale trends. We also present two alternative models, a regression model with ARMA errors and a Tweedie model, which we use to assess the robustness of our results to model misspecification. In Section 4, we study the ability of our models to predict future rainfall in comparison to naive methods based on monthly sample means or medians. We conclude with a discussion in Section 5.

## 2. Station Data and Preliminary Analyses

The purpose of this section is to describe the data used in the subsequent analyses and to illustrate the MSD.

Earlier work [19] describes how the longest possible sequences of daily rainfall totals for as many stations as possible in the study region were sought from *Instituto Meteorológico Nacional* (IMN), the National Meteorological Institute of Costa Rica. These authors selected data from Nicoya, Paquera, Garza, La Guinea, and Liberia (see Figure 1 for station locations), filling in missing data where possible to facilitate inter-station analysis.

A simple view of spatial rainfall structure across the study domain was derived from inter-station correlation analyses. Correlation coefficient matrices based on daily

and weekly rainfall showed that rainfall amounts at each pair of stations are only weakly correlated on these shorter time scales. In contrast, monthly totals are highly correlated. Therefore, these monthly values were used to demonstrate the annual rainfall pattern, namely, the wet and dry seasons and the MSD, as depicted in Figure 2 [19].

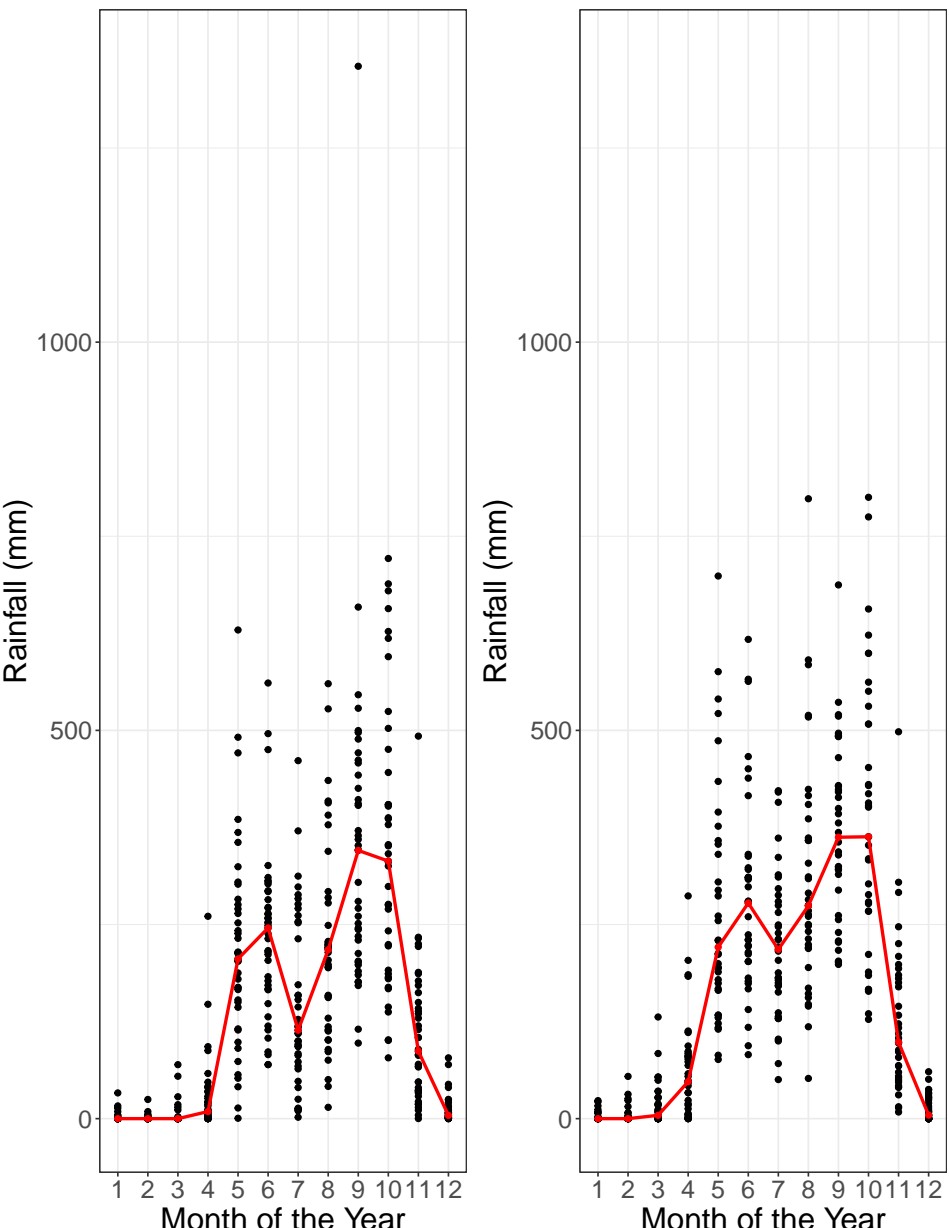

**Figure 2.** Annual cycle of total monthly rainfall values at the Liberia station (**left**) from 1980 to 2020 and at the Nicoya station (**right**) from 1980 to 2016. The red line represents the observed median rainfall in each month.

Due to their very weak spatial structure and long sequences of observed values, in the present paper, we focus on time series of monthly total rainfall for only the Liberia and Nicoya stations.

Nicoya is located at the crest of the ridge running parallel to the shoreline down the spine of the Nicoya Peninsula, and Liberia is in the lowlands between the peninsula and the major mountain range making up the Costa Rican Highlands. Choosing these two stations affords an opportunity to detect any effects of local topographic rainfall forcing embedded in larger-scale effects.

The literature provides a strong indication that rainfall in this region is influenced by ENSO (e.g., [20]). We accessed monthly ONI data from the National Weather Service Climate Prediction Center website. We used these data to estimate the influences of ENSO on the various features of the annual cycle of rainfall at Liberia and Nicoya.

Of considerable local interest is the possibility that Guanacaste rainfall is subject to longer term trends, possibly due to changing climates. The data sequences observed from 1980 to 2020 (Liberia) and from 1980 to 2016 (Nicoya) provide the opportunity to detect such secular trends, though the high degree of inter-annual variability makes this task difficult [19].

## 3. Statistical Models and Estimation of the ONI Effect and Long-Term Time Trend

In this section, we develop the double-Gaussian model, a parametric model that we use to describe the annual cycle in the rainfall observations, and estimate its parameters. We discuss the advantages and disadvantages of this model. We also present two alternative models that describe the annual cycle semi-parametrically (by allowing a different parameter to characterize each monthly mean): a classical regression model with autoregressive moving average (ARMA) errors and a model based on the Tweedie distribution and other more realistic assumptions. These alternative models provide a means of assessing the sensitivity of our results to the assumptions inherent in the double-Gaussian model. We fit these three models to the data from Liberia and Nicoya. We interpret the resulting parameter estimates, paying particular attention to the estimates of the ONI effect and long-term time trend.

We first define notation common to all three models. In particular, let $Y_{it}$ be the total rainfall (in mm) in month $t$ of year $i$, $t = 1, \ldots, 12$, $i = 1, \ldots, 36$. Let $x_{it1}$ be ONI (in $^\circ$C) in month $t$ of year $i$. Let time be represented by $x_{it2}$, the number of months between our starting date, December 1979, and month $t$ of year $i$. We also define the constant $n$ as the total number of months over which rainfall was observed ($n = 492$ for Liberia and $n = 443$ for Nicoya). In all models, we incorporate time via the predictor $x_{it2}/n$ (rather than $x_{it2}$) because, in some contexts (e.g., [21]), the estimated coefficient of a predictor that grows with $n$ is inconsistent.

The Nicoya series includes five missing values, which we assume are missing completely at random and are therefore excluded.

### 3.1. Double-Gaussian Model

The double-Gaussian model, which we present in this section, uses a simple, parametric, functional form to approximate the two-peak, annual rainfall pattern that is so strongly evident in Figure 2. Let $Y_{it}^* = Y_{it}/(1 \text{ mm})$, a unitless measure of total rainfall in month $t$ of year $i$. Our model assumes that

$$Z_{it} \equiv \ln(Y_{it}^* + 1) \sim N(\ln(\lambda_{it}), \sigma_t^2) \tag{1}$$

where

$$\lambda_{it} = A_1 \exp\left\{-\frac{(t - l_1)^2}{\phi_1}\right\} + A_2 \exp\left\{-\frac{(t - l_2)^2}{\phi_2}\right\} + \delta. \tag{2}$$

In other words, $\lambda_{it}$ is a linear combination of two Gaussian-shaped basis functions. Here, $A_1$ and $A_2$ are the amplitudes of the median spring and summer peaks, respectively. The two peaks are centered at locations (months) $l_1$ and $l_2$, with widths controlled by $\phi_1$ and $\phi_2$ (months squared). The parameter $\delta$ is an offset.

We allow the amplitudes of the peaks to depend on ONI—and extended their model by allowing the amplitudes to depend on time as well. In our version, we let $A_1 = \exp\{\beta_{10} + \beta_{11}x_{it1} + \beta_{12}\frac{x_{it2}}{n}\}$ and $A_2 = \exp\{\beta_{20} + \beta_{21}x_{it1} + \beta_{22}\frac{x_{it2}}{n}\}$. Since the variability of rainfall is lower in the wet months (May–October), we define $\sigma_t \equiv \sigma_1$ for $5 \le t \le 10$ and $\sigma_t \equiv \sigma_0$ otherwise.

Although the model is expressed on the logarithmic scale (to stabilize the variance of the rainfall observations), the parameter $\lambda_{it}$ has a ready interpretation on the original scale as the median of $Y_{it} + 1$. (We added 1 mm to each rainfall measurement as a way of handling zeroes in the data.)

Note that "Gaussian" in the name "double-Gaussian model" refers to the *shape* of the two Gaussian basis functions in (2). In contrast, $Z_{it}$ is assumed to have a Gaussian *distribution*. We emphasize that this model is not equivalent to a 2-component Gaussian mixture model; (2) describes the annual cycle of rainfall over the course of one year, not the probability distribution of rainfall in a given month.

The double-Gaussian model relies on one further critical assumption: the $Y_{it}$s are treated as independent. We assessed the reasonableness of this and the other model assumptions after fitting the model.

Since we estimated the model parameters using maximum likelihood estimation and prefer to avoid constrained optimization methods, we reparameterized the model so that the range of every parameter is the entire real line. In particular, we defined $\phi_1^* = \log \phi_1$, $\phi_2^* = \log \phi_2$, $\delta^* = \log \delta$, $\sigma_0^* = \log \sigma_0$, and $\sigma_1^* = \log \sigma_1$. We then estimated the parameters $(\beta_{10}, \beta_{11}, \beta_{12}, l_1, \phi_1^*, \beta_{20}, \beta_{21}, \beta_{22}, l_2, \phi_2^*, \delta^*, \sigma_0^*, \sigma_1^*)$.

The maximum likelihood estimates of the parameters and their standard errors (for each station separately) are listed in Table 1. Of particular interest in our analysis are $\beta_{11}$ and $\beta_{21}$, the parameters that represent the effects of ONI (in $°C^{-1}$), and $\beta_{12}$ and $\beta_{22}$, the parameters that represent the (unitless) effect of time. For both stations, we have evidence at the 5% level that ONI is negatively associated with spring rainfall (i.e., that $\beta_{11}$ is negative). For a 1 $°C$ increase in ONI, the amplitude of the spring peak changes by an estimated factor of 0.558 in Liberia and 0.805 in Nicoya. Likewise, for both stations, we have evidence that ONI is negatively associated with summer rainfall (i.e., that $\beta_{21}$ is negative); for a 1 $°C$ increase in ONI, the amplitude of the summer peak changes by an estimated factor of 0.757 in Liberia and 0.811 in Nicoya. Likelihood ratio tests provide evidence that the effect of ONI on the spring peak is different from that on the summer peak in Liberia (*p*-value = 0.026), but there is no such evidence for Nicoya (*p*-value = 0.934). The data provide no evidence of a long-term time trend in rainfall at either station.

The primary advantage of the double-Gaussian model is the parametric form of the median rainfall, which provides a clear description of the annual cycle. This structure allows easy quantification of the annual cycle of rainfall for hydrologic model input.

Possible weaknesses of the double-Gaussian model are the normality and independence assumptions. Diagnostic plots of the standardized residuals (included in the online supplementary material) show that the normality assumption is violated in the drier months, where many zero values occur. However, normality is often unimportant for the validity of likelihood-based inference procedures. With respect to the independence assumption, the estimated autocorrelation functions of the standardized residuals (depicted in the online supplementary material) provide no suggestion of autocorrelation in the residuals. The assumption of independence of the rainfall observations is thus reasonable, despite their time-series nature. The explanation is that the ONI values effectively explain the autocorrelation in the rainfall measurements. Therefore, while the model provides a good description of the data, its predictive capabilities are restricted to the case where future values of ONI or other similar variables can be treated as known or can be forecast a few months ahead (as is the case for ONI, as discussed in Section 4).

Given these considerations, we cautiously take our inference based on the double-Gaussian model as valid. However, to assess the robustness of our conclusions to model misspecification, we compared our findings to those based on two different models, the regression model with ARMA errors and the Tweedie model.

**Table 1.** Estimates and standard errors of the parameters of the double-Gaussian model.

| Station | Parameter | Estimate | Standard Error | *p*-Value |
|---|---|---|---|---|
| Liberia | $\beta_{10}$ | 5.774 | 0.150 | 0.000 |
| | $\beta_{11}$ | −0.583 | 0.117 | 0.000 |
| | $\beta_{12}$ | −0.342 | 0.237 | 0.150 |
| | $l_1$ | 5.820 | 0.050 | 0.000 |
| | $\phi_1^*$ | −0.021 | 0.083 | 0.798 |
| | $\beta_{20}$ | 5.849 | 0.128 | 0.000 |
| | $\beta_{21}$ | −0.278 | 0.065 | 0.000 |
| | $\beta_{22}$ | 0.185 | 0.206 | 0.368 |
| | $l_2$ | 9.198 | 0.062 | 0.000 |
| | $\phi_2^*$ | 0.593 | 0.074 | 0.000 |
| | $\delta^*$ | 0.426 | 0.105 | 0.000 |
| | $\sigma_0^*$ | 0.143 | - | - |
| | $\sigma_1^*$ | −0.330 | - | - |
| Nicoya | $\beta_{10}$ | 5.825 | 0.099 | 0.000 |
| | $\beta_{11}$ | −0.217 | 0.077 | 0.005 |
| | $\beta_{12}$ | −0.237 | 0.163 | 0.144 |
| | $l_1$ | 5.885 | 0.071 | 0.000 |
| | $\phi_1^*$ | 0.621 | 0.096 | 0.000 |
| | $\beta_{20}$ | 6.054 | 0.094 | 0.000 |
| | $\beta_{21}$ | −0.209 | 0.048 | 0.000 |
| | $\beta_{22}$ | 0.093 | 0.154 | 0.544 |
| | $l_2$ | 9.137 | 0.060 | 0.000 |
| | $\phi_2^*$ | 0.652 | 0.079 | 0.000 |
| | $\delta^*$ | 0.594 | 0.166 | 0.000 |
| | $\sigma_0^*$ | 0.343 | - | - |
| | $\sigma_1^*$ | −0.782 | - | - |

*3.2. Regression Model with ARMA Errors*

The regression model with autoregressive moving-average (ARMA) errors is a standard model for describing autocorrelated, normally distributed outcomes whose means depend on predictor variables (e.g., [22,23]). It is therefore a reasonable alternative to the double-Gaussian model. As explained in the online supplementary material, we consider, in particular, the regression model with AR(1) errors applied to a transformation of the rainfall measurements, $Y_{it}^{1/4}$. We use standard diagnostic tools such as Q-Q plots and autocorrelation function plots of residuals as justification for this choice (e.g., [23]).

Our model can be expressed as

$$Y_{it}^{1/4} = m_t + \gamma_1 \delta_{t1} x_{it1} + \gamma_2 \delta_{t2} x_{it1} + \gamma_3 \delta_{t3} x_{it1} + \gamma_4 \frac{x_{it2}}{n} + \epsilon_{it}, \qquad (3)$$

where $m_t$ is the effect of month $t$ (in $mm^{1/4}$). The variable $\delta_{t1}$ is one if $t < 5$ or $t > 10$ (i.e., to indicate the dry season) and zero otherwise. The variable $\delta_{t2}$ is one if $5 \leq t \leq 7$ (i.e., to indicate the spring peak in May, June, and July) and zero otherwise. The variable $\delta_{t3}$ is one if $8 \leq t \leq 10$ (i.e., to indicate the summer peak in August, September, and October) and zero otherwise. The purpose of the variables $\delta_{t1}$, $\delta_{t2}$, and $\delta_{t3}$ is to allow for an interaction effect between month and ONI; in particular, the effect of ONI (in units of $mm^{1/4}(°C)^{-1}$ is assumed to be $\gamma_1$ during the dry months, $\gamma_2$ during the spring peak, and $\gamma_3$ during the summer peak. (With a larger sample size, the model could be generalized to allow an ONI by month—rather than an ONI by season—interaction.) The parameter $\gamma_4$ represents the effect of time (in $mm^{1/4}$). Finally, $\epsilon_{it}$ follows an AR(1) process.

We fit the regression model with AR(1) errors using the `arima` function in the R package `stats`. Table 2 shows the estimates of some of the parameters and their standard errors. (The remainder appear in the online supplementary material.) Of note are the statistically significant effects of ONI on the spring peak (Liberia) and summer peaks (Liberia and Nicoya). However, we found no evidence of a long-term time trend. A caveat

to our findings is that we used the same dataset both to select the model and to make inferences about the regression coefficients. Consequently, the reported p-values may be too small. The estimated lag-1 coefficients are small for both stations. In other words, after adjusting for month and ONI effects, the rainfall observations are approximately independent. These results are broadly consistent with those based on the double-Gaussian model, lending strength to our conclusions based on the latter and, importantly, further justifying the independence assumption.

**Table 2.** Estimates and standard errors of some of the parameters of the regression model with AR(1) errors.

| Station | Parameter | Estimate | Standard Error | *p*-Value |
|---------|-----------|----------|----------------|-----------|
| Liberia | $\gamma_1$ | −0.104 | 0.049 | 0.032 |
| | $\gamma_2$ | −0.380 | 0.108 | 0.000 |
| | $\gamma_3$ | −0.303 | 0.074 | 0.000 |
| | $\gamma_4$ | 0.014 | 0.115 | 0.902 |
| | Lag 1 Coefficient | 0.076 | 0.045 | - |
| | Residual Variance | 0.459 | - | - |
| Nicoya | $\gamma_1$ | −0.025 | 0.055 | 0.646 |
| | $\gamma_2$ | −0.152 | 0.121 | 0.212 |
| | $\gamma_3$ | −0.239 | 0.083 | 0.004 |
| | $\gamma_4$ | −0.105 | 0.136 | 0.441 |
| | Lag 1 Coefficient | 0.100 | 0.048 | - |
| | Residual Variance | 0.538 | - | - |

The model with ARMA errors provides a flexible way of handling autocorrelation in the rainfall measurements—a key advantage relative to the double-Gaussian model. In particular, the double-Gaussian model requires the inclusion of a predictor variable (such as ONI) to explain the autocorrelation in the observation. On the other hand, the model with ARMA errors does not have this requirement because it can explain the autocorrelation via the error terms. It therefore allows rainfall to be predicted even when future values of such predictor variables are unknown.

However, this model has three major drawbacks in our setting. First, like the double-Gaussian model, it is based on the assumption of normality, which, according to QQ plots of the residuals (not shown), is clearly unrealistic in the dry months. Second, interpretation of the effects of predictors is challenging given that they are specified on the scale of the transformed response (and neither the mean nor the median of the untransformed response has a simple relationship with the predictors). Third, while the non-parametric specification of the annual cycle is a flexible approach, it lacks the descriptive appeal of the double-Gaussian model.

### 3.3. Tweedie Model

To address our concerns about the normality assumption required of the double-Gaussian model and regression model with AR(1) errors, we also consider a Tweedie model [24]. The Tweedie distribution (e.g., [25]) does not, in general, have a closed form but is flexible and includes the gamma and compound Poisson-gamma distributions as special cases. The latter is a continuous distribution with a point mass at zero—a feature that is especially appealing in our context, where certain months regularly see zero rainfall.

To be more specific, given month-specific random effects, we assume that monthly rainfall follows a Tweedie distribution with a mean depending on the month of the year, ONI, and time (in number of months since December 1979). The random effects, which are autocorrelated, induce autocorrelation in the monthly rainfall observations. As in the regression model with AR(1) errors, we specify the mean structure non-parametrically, with one parameter for each month of the year.

The specification of the model is as follows. Again, let $Y_{it}^* = Y_{it}/(1\ mm)$, a unitless measure of total rainfall in month $t$ of year $i$. Let $\mu_{it}$ be the expected value of $Y_{it}^*$. We model $\mu_{it}$ as

$$\mu_{it} = \exp\left(a_t + b_1\delta_{t1}x_{it1} + b_2\delta_{t2}x_{it1} + b_3\delta_{t3}x_{it1} + b_4\frac{x_{it2}}{n}\right), \tag{4}$$

where $a_t$ is the (unitless) effect of month $t$; $\delta_{tk}$, $k = 1, 2, 3$ are defined as in Section 3.2; $b_1$, $b_2$, and $b_3$ are the effects of ONI (in units of $°C^{-1}$) during the dry months, spring peak, and summer peak, respectively; and $b_4$ is the (unitless) effect of time.

In addition, let $U_{it} \geq 0$ be a unitless, unobserved random effect associated with month $t$ of year $i$. In our context, these random effects are not assumed to have a physical interpretation (though they may well correspond to the local and remote meteorological processes that affect rainfall, as outlined in Section 1). Rather, they are included as a convenient means of allowing for autocorrelation in the observed rainfall totals. We assume that, given the random effects, the $Y_{it}^*$s are independent and Tweedie-distributed. We further assume that

$$
\begin{aligned}
\mathrm{E}[Y_{it}^* \mid U_{it}] &= \mu_{it}U_{it} \\
\mathrm{Var}[Y_{it}^* \mid U_{it}] &= \phi_t\mu_{it}^{q_t}U_{it} \\
\mathrm{E}[U_{it}] &= 1 \\
\mathrm{Var}[U_{it}] &= \tau^2 \\
\mathrm{Cov}[U_{is}, U_{jt}] &= \tau^2\rho^{d_{isjt}},
\end{aligned}
$$

where $\phi_t$, $q_t$, and $\rho$ are unknown, unitless parameters to be estimated; $0 \leq \rho < 1$; $t > s$; and $d_{isjt} = 12(j - i) + t - s > 0$ represents the number of months between observations $(i, s)$ and $(j, t)$ (divided by 1 month to preserve dimensional homogeneity). These assumptions imply that $\mathrm{Cov}[Y_{is}^*, Y_{jt}^*] = \mu_{is}\mu_{jt}\tau^2\rho^{d_{isjt}}$—i.e., that $Y_{is}^*$ and $Y_{jt}^*$ are correlated; the correlation decreases to zero as $d_{isjt} \to \infty$. Our model builds on earlier work [24], allowing $\phi_t$ and $q_t$ to depend on month.

Previous authors [24] estimated most of the parameters in the Tweedie model using an iterative procedure. In particular, at each iteration, they replaced the random effects with their best unbiased predictors, estimated the regression parameters using unbiased estimating equations, and estimated the remaining parameters via the method of moments (MOM; e.g., [26]). They repeated this procedure until convergence was achieved. We used the same method of estimation (with an adjustment to the MOM estimators since $\phi_t$ can differ by month in our model). Rather than the suggested Newton scoring algorithm, we used the `multiroot` function in the R package `rootSolve` to solve the estimating equations. For simplicity, we handle the missing values in the Nicoya data by replacing them with the sample median rainfall for their corresponding months. Following previous authors [24], we did not estimate $q_t$. Instead, for $t = 1, 2, 3, 4$, we set $q_t = 1.62$ (the maximum likelihood estimate obtained by assuming independent Tweedie observations in those months). For $t = 5, 6, \ldots, 12$, we set $q_t = 2$, which corresponds to the gamma distribution. This distribution provides a reasonable fit to the rainfall in those months (where no zero rainfall observations occurred). The online supplementary material contains details about the goodness of fit of this model.

Table 3 shows the resulting estimates of some of the parameters and their standard errors. We have no evidence of an ONI effect in the dry months ($p = 0.967$ in Liberia and $p = 0.732$ in Nicoya) or of a time trend ($p = 0.738$ in Liberia and $p = 0.069$ in Nicoya). However, the effects of ONI on the spring and summer peaks (i.e., $b_2$ and $b_3$) are significant at both stations. In particular, if ONI in month $t$ of year $i$ is 1 $°C$ greater than in month $t$ of year $j$ and $t$ is a spring month, then the expected rainfall in month $t$ of year $i$ is estimated to differ from that in month $t$ of year $j$ by a factor of $e^{-0.281} = 0.755$ (95% confidence interval: $[0.634, 0.900]$) in Liberia and $e^{-0.147} = 0.863$ (95% confidence interval: $[0.752, 0.991]$) in Nicoya. Similarly, if ONI in month $t$ of year $i$ is 1 $°C$ greater than in month $t$ of year $j$ and $t$ is a summer month, then the expected rainfall in month $t$ of year $i$ is estimated to

differ from that in month $t$ of year $j$ by a factor of $e^{-0.322} = 0.725$ (95% confidence interval: $[0.659, 0.797]$) in Liberia and $e^{-0.231} = 0.794$ (95% confidence interval: $[0.733, 0.859]$) in Nicoya. These findings, like those from the regression model with AR(1) errors, reinforce our conclusions based on the double-Gaussian model. They also suggest that the impact of the violation of the normality assumption in the double-Gaussian model may be minimal.

**Table 3.** Estimates and standard errors of some of the parameters of the Tweedie model.

| Station | Parameter | Units | Estimate | Standard Error | *p*-Value |
|---------|-----------|-------|----------|----------------|-----------|
|         | $b_1$     | $°C^{-1}$ | −0.004 | 0.097 | 0.967 |
|         | $b_2$     | $°C^{-1}$ | −0.281 | 0.089 | 0.002 |
| Liberia | $b_3$     | $°C^{-1}$ | −0.322 | 0.048 | <0.001 |
|         | $b_4$     | $°C^{-1}$ | 0.037  | 0.111 | 0.738 |
|         | $\tau^2$  | -     | 0.024    | -     | -     |
|         | $\rho$    | -     | 0.437    | -     | -     |
|         | $b_1$     | $°C^{-1}$ | −0.029 | 0.086 | 0.732 |
|         | $b_2$     | $°C^{-1}$ | −0.147 | 0.070 | 0.036 |
| Nicoya  | $b_3$     | $°C^{-1}$ | −0.231 | 0.040 | <0.001 |
|         | $b_4$     | $°C^{-1}$ | −0.173 | 0.095 | 0.069 |
|         | $\tau^2$  | -     | 0.022    | -     | -     |
|         | $\rho$    | -     | 0.467    | -     | -     |

As in the analysis based on the regression model with AR(1) errors, after accounting for month and ONI effects, the estimated residual autocorrelation was very low (maximum estimated lag-1 correlation of 0.055 for Liberia and 0.087 for Nicoya).

The Tweedie model has a number of advantages. First, as mentioned earlier, it provides an excellent approximation of the distribution of rainfall on the original scale, capturing both the zero observations and the (approximately continuous) non-zero observations. A second appealing feature is the model's high degree of interpretability. In particular, the effects of the predictor variables have a simple interpretation as (multiplicative) changes in mean monthly rainfall. Finally, like the regression model with AR(1) errors, the Tweedie model can incorporate autocorrelation in the responses and can therefore be used to model rainfall with or without the inclusion of ONI as a predictor.

The main weakness of the Tweedie model—shared by the AR(1) model—is its non-parametric treatment of the month effects, which is undesirable from the perspective of description. In addition, estimation of the parameters of this model requires customized code and non-trivial computing time, whereas estimation in the context of the double-Gaussian model and the regression model with AR(1) errors can be performed using standard statistical software and requires mere seconds.

### 3.4. Summary of Models and Inferential Results

Our models have advantages and disadvantages. Of the three models, the double-Gaussian model provides the simplest, most interpretable description of rainfall in the region. However, its simple structure leads to limitations from a prediction standpoint—namely, future ONI values must be known with a high degree of certainty. The regression model with AR(1) errors and the Tweedie model overcome this limitation but rely on less interpretable, semi-parametric mean structures. Another advantage of the Tweedie model is that its assumptions are more statistically defensible than those of other two models (leading to more reliable inference), but this advantage comes at the cost of increased computational effort required to estimate parameters.

While a comparison of effect estimates across models would not be sensible given that they are on different scales, we can say that inference based on all three models leads to the same general conclusions: evidence of a negative impact of ONI on spring and summer rainfall and no evidence of a time trend.

## 4. Rainfall Forecasting

In this section, we discuss the use of our models for forecasting future rainfall. Due to the inherent variability of rainfall (see Figure 2), we do not expect any of our models to be able to predict rainfall precisely. Rather, our question of interest is whether our models provide an improvement over naive prediction methods based on monthly sample mean or median rainfall.

Our focus is point prediction rather than formal inference; therefore, the points we raised in Section 3 regarding the validity of the assumptions of some of our models are less of a concern. Prediction intervals, which are more heavily dependent on the validity of the model assumptions, are beyond the scope of this paper, but we discuss them in general terms in Section 5.

We are interested in prediction in two cases: the case where future values of ONI are treated as unknown and not used in modelling and the case where future values of ONI are routinely predicted (e.g., by the Climate Prediction Center) and thus may be treated as known. We omit time as a predictor in all cases given its earlier demonstrated insignificance.

To assess the predictive ability of our proposed models, we forecast the following rainfall summary quantities over the years 2006–2020 in Liberia and 2007–2016 in Nicoya:

1. Spring peak (total rainfall in May, June, and July) conditional on all observations prior to January of the year in question.
2. Summer peak (total rainfall in August, September and October) conditional on all observations prior to January of the year in question.
3. Annual total conditional on all observations prior to January of the year in question.
4. Monthly rainfall conditional on all observations prior to the month in question ("One-month-ahead predictions").
5. Monthly rainfall conditional on all but the last three observations prior to the month in question ("Three-month-ahead predictions").

Of particular interest are the predictions of the spring and summer peaks because of their importance for local, rain-fed agriculture and the predictions of annual total rainfall because of their importance for groundwater recharge and hydroelectricity production.

Comparing the predictive ability of the three models is challenging because they are designed to predict fundamentally different quantities. In particular, the double-Gaussian model is designed to predict mean monthly total rainfall on the $\log(Y_{it} + 1)$ scale given future ONI values; the AR(1) model is designed to predict mean monthly total rainfall on the $Y_{it}^{0.25}$ scale with or without knowledge of future ONI values; and the Tweedie model is designed to predict mean monthly total rainfall on the original scale with or without knowledge of future ONI values. Since the scales differ across models, we cannot compare their monthly predictions directly. The only exception is the case of the double-Gaussian model and the AR(1) model; the back-transformation of their predictions results in predicted median monthly rainfall in both cases.

In the same vein, predicted mean spring, summer, and annual rainfall, obtained by summing the relevant monthly predictions, are on three different scales. Those based on the Tweedie model are on the original scale and are thus readily interpretable; those based on the other two models are on transformed scales and cannot be back-transformed to obtain predicted median totals. Therefore, we do not compare the predictions of mean spring, summer, and annual rainfall across models directly.

However, we can derive "naive predictions" of rainfall on any scale over any subset of months based on the appropriate sample means or medians from our data. Thinking of the naive predictions as a benchmark, we can then observe whether our methods result in improved predictions. For example, we can compare the estimated mean spring rainfall on the $\log(Y_{it} + 1)$ scale based on the double-Gaussian model to the sample mean spring rainfall on the same scale.

To make predictions using our models, we require estimates of their parameters. As fitting the Tweedie model involves some computational time, when predicting monthly rainfall (quantities 4 and 5 above), we used the parameter estimates resulting from fitting

the model to all data observed prior to January of the year in question. In contrast, fitting the regression model with AR(1) errors or the double-Gaussian model is essentially instantaneous. Thus, for predictions based on these models, we used parameter estimates derived by fitting the model to all data observed prior to the time in question. Importantly, in all cases, only *past* data were used to predict future rainfall.

When comparing estimated means, we used mean squared error (MSE) to compare the two sets of predictions (since mean rainfall is the optimal predictor when MSE is the chosen loss function); when comparing estimated medians, we used mean absolute error (MAE) as a measure of prediction error (since the median is the optimal predictor when MAE is the chosen loss function).

The missing data from the Nicoya station were omitted from all predictions.

Let $\mathbf{Y} = (Y_{11}, Y_{12}, \ldots, Y_{IT})$ be the random vector of rainfall measurements observed up to year $I$, month $T$, and let $\mathbf{y}$ be the observed realization of this random vector. Let $Y_{new}$ be a new monthly total that we wish to predict.

*4.1. Double-Gaussian Model*

For the double-Gaussian model, we use $\mathbf{y}$ to find the maximum likelihood estimates of the parameters in (1) and (2). Using these parameter estimates and the month and ONI value corresponding to $Y_{new}$, we can then estimate $\lambda_{new} = \mathrm{E}[\log(Y_{new} + 1)]$. We use $\hat{\lambda}_{new}$ as the point predictor of $\mathrm{E}[\log(Y_{new} + 1)]$, and we use $\exp(\hat{\lambda}_{new})$ as the point predictor of the median of $Y_{new} + 1$.

Table 4 shows the ratios of the prediction errors based on the double-Gaussian model (where future ONI values are assumed known) to those based on the appropriate naive method (sample mean or sample median rainfall over the specified subset of months). For spring, summer, and annual totals, the prediction errors based on the double-Gaussian model are substantially smaller than those based on the naive method; the reduction in error is 17–27% in the case of the spring totals, 35–41% in the case of the summer totals, and 40–49% in the case of the annual totals. However, for 1- and 3-month-ahead predictions, the two methods are approximately equivalent. Figures 3–9 provide a different view of these results, displaying the observed and predicted values over the various time scales.

**Table 4.** Ratio of the prediction error (PE) based on the double-Gaussian model (where ONI is assumed known) to that based on the naive method (sample means/medians by month).

| Station | Quantity | Scale | Mean or Median | Error Measure | PE Ratio |
|---------|----------|-------|----------------|---------------|----------|
| Liberia | Spring total | $\log(Y_{it} + 1)$ | Mean | MSE | 0.730 |
| | Summer total | $\log(Y_{it} + 1)$ | Mean | MSE | 0.591 |
| | Annual total | $\log(Y_{it} + 1)$ | Mean | MSE | 0.608 |
| | 1-month-ahead | $Y_{it}$ | Median | MAE | 1.050 |
| | 3-month-ahead | $Y_{it}$ | Median | MAE | 1.049 |
| Nicoya | Spring total | $\log(Y_{it} + 1)$ | Mean | MSE | 0.836 |
| | Summer total | $\log(Y_{it} + 1)$ | Mean | MSE | 0.655 |
| | Annual total | $\log(Y_{it} + 1)$ | Mean | MSE | 0.515 |
| | 1-month-ahead | $Y_{it}$ | Median | MAE | 1.003 |
| | 3-month-ahead | $Y_{it}$ | Median | MAE | 0.999 |

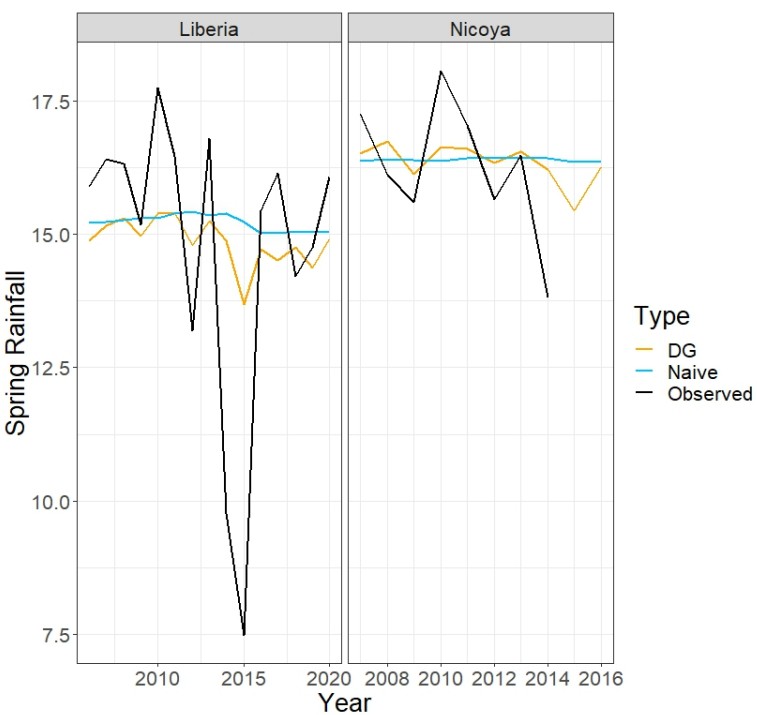

**Figure 3.** Predicted (using the double Gaussian and naive approaches) and observed total spring rainfall values at the Liberia station (**left**) from 2006 to 2020 and at the Nicoya station (**right**) from 2007 to 2016. Rainfall is reported on the scale $\log(Y_{it}) + 1$.

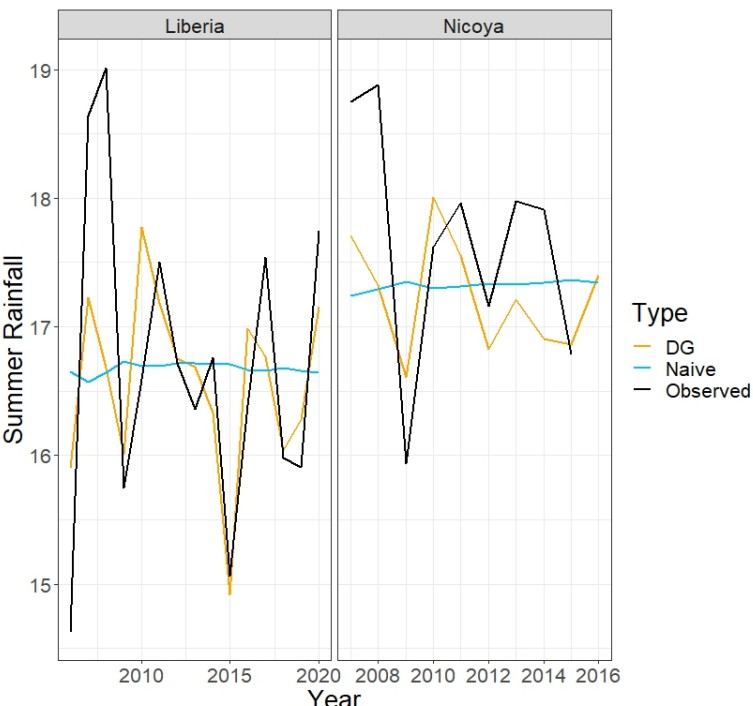

**Figure 4.** Predicted (using the double Gaussian and naive approaches) and observed total summer rainfall values at the Liberia station (**left**) from 2006 to 2020 and at the Nicoya station (**right**) from 2007 to 2016. Rainfall is reported on the scale $\log(Y_{it}) + 1$.

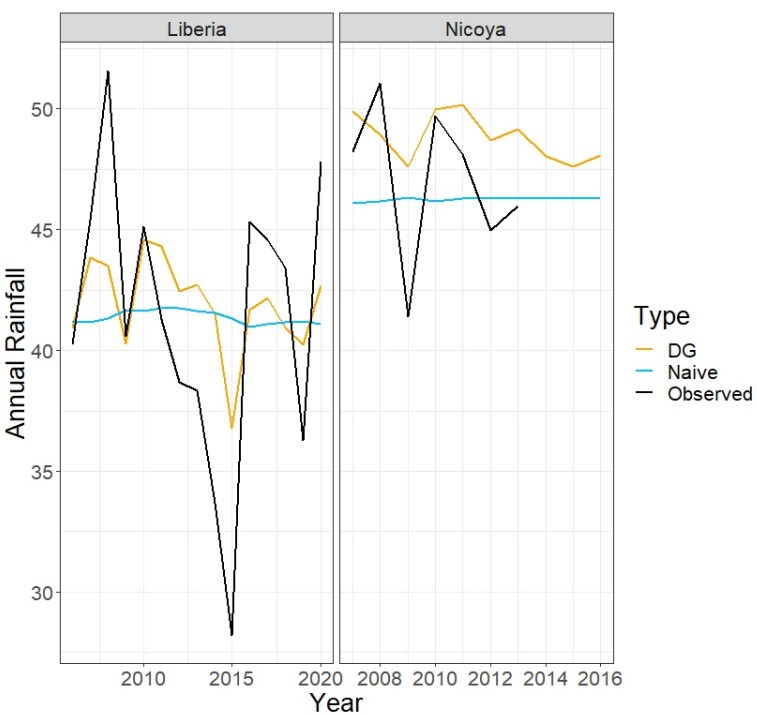

**Figure 5.** Predicted (using the double Gaussian and naive approaches) and observed total annual rainfall values at the Liberia station (**left**) from 2006 to 2020 and at the Nicoya station (**right**) from 2007 to 2016. Rainfall is reported on the scale $\log(Y_{it}) + 1$.

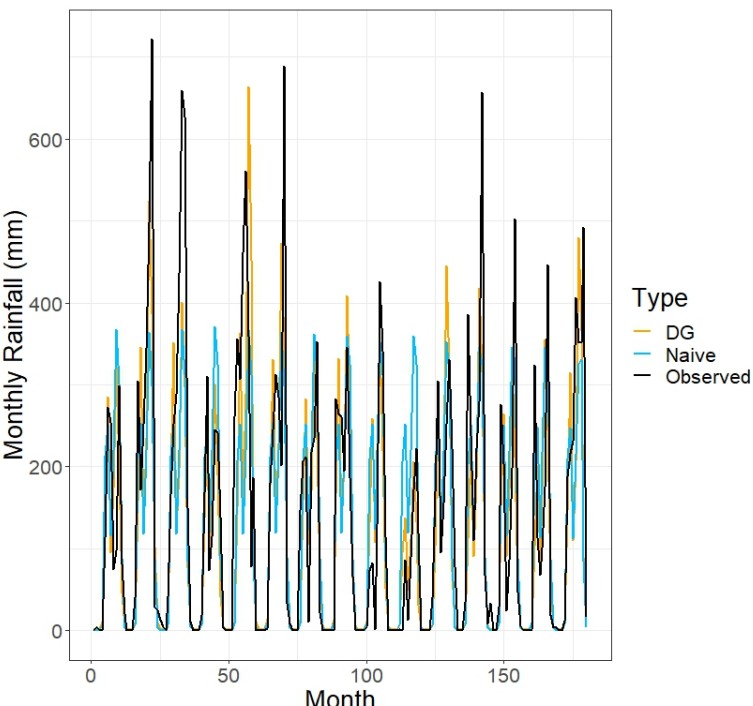

**Figure 6.** One-month-ahead predicted (using the double Gaussian and naive approaches) and observed monthly rainfall values at the Liberia station from 2006 to 2020. "Month" is the number of months since January 2006.

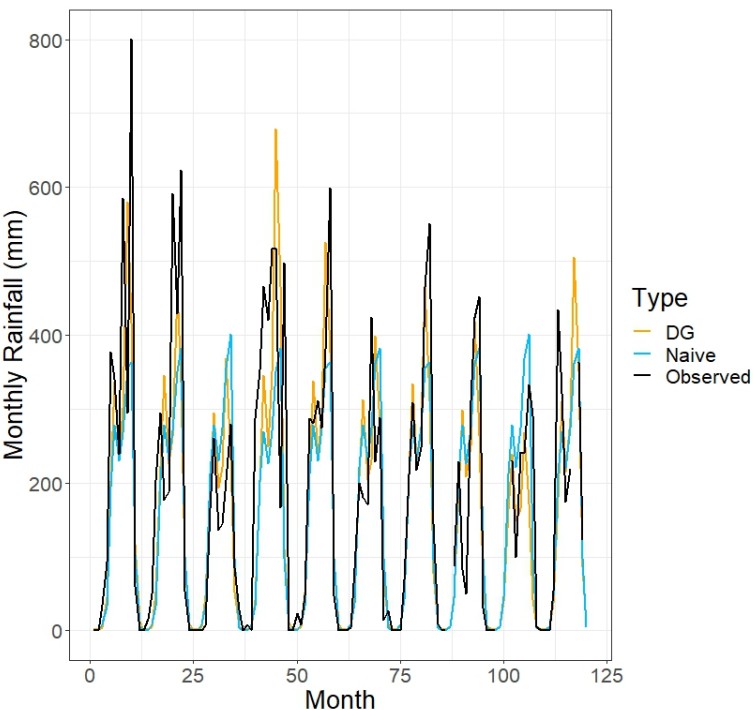

**Figure 7.** One-month-ahead predicted (using the double Gaussian and naive approaches) and observed monthly rainfall values at the Nicoya station from 2007 to 2016. "Month" is the number of months since January 2007.

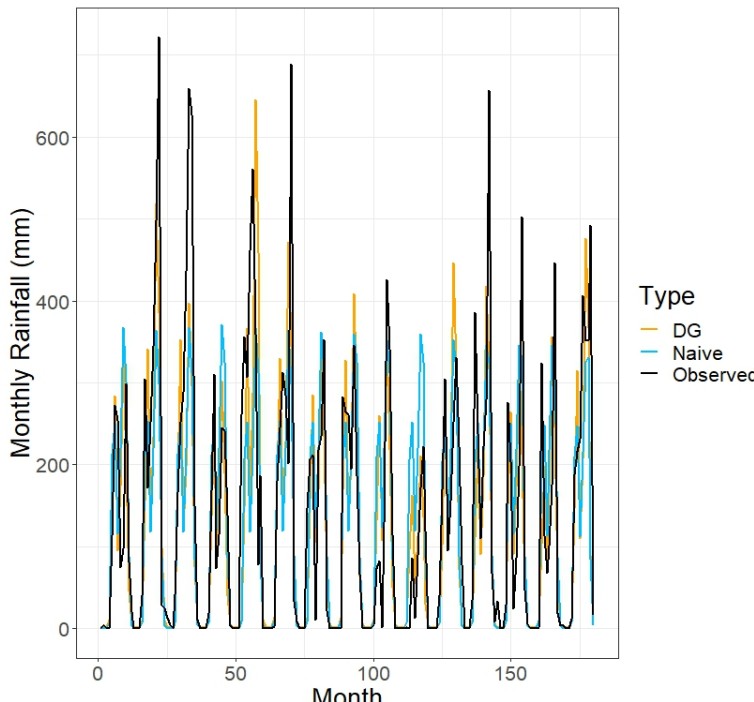

**Figure 8.** Three-month-ahead predicted (using the double Gaussian and naive approaches) and observed monthly rainfall values at the Liberia station from 2006 to 2020. "Month" is the number of months since January 2006.

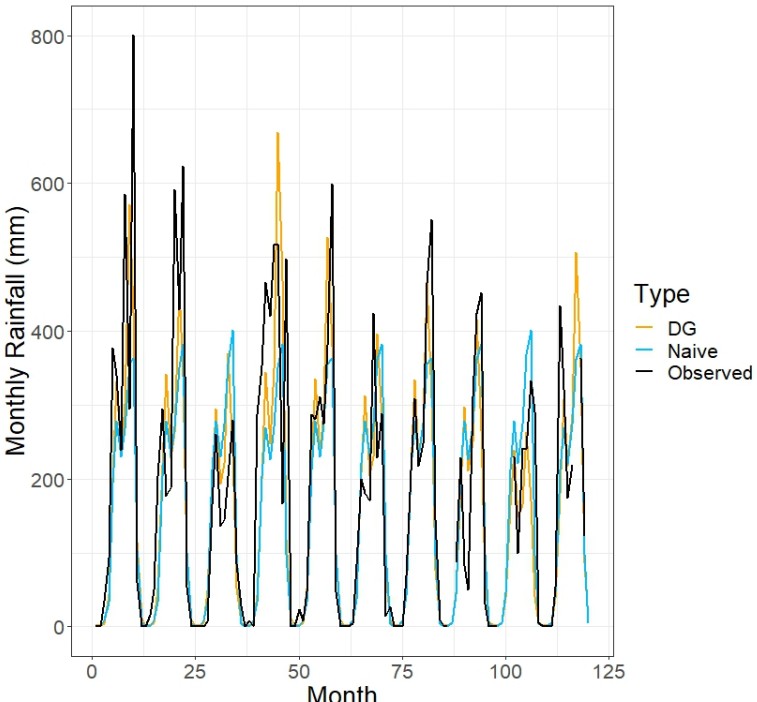

**Figure 9.** Three-month-ahead predicted (using the double Gaussian and naive approaches) and observed monthly rainfall values at the Nicoya station from 2007 to 2016. "Month" is the number of months since January 2007.

*4.2. Regression Model with AR(1) Errors*

For the regression model with AR(1) errors, we first define

$$\hat{\eta} = \hat{\mathrm{E}}\left[Y_{new}^{0.25}\big|\mathbf{Y} = \mathbf{y}\right],$$

which we calculate using the model fit with or without ONI, as desired, and the month and ONI value (if applicable) corresponding to $Y_{new}$. Given past rainfall observations $\mathbf{y}$, we use $\hat{\eta}$ as a point predictor of the mean of $Y_{new}^{0.25}$, and we use $\hat{\eta}^4$ as a point predictor of the median of $Y_{new}$. We use this method (via the R function `forecast` in the package `forecast`) to predict rainfall when the future ONI is treated as known or unknown.

Table 5 provides the ratios of the prediction errors based on the regression model with AR(1) errors (where future ONI values may be assumed known or unknown) to that based on the appropriate naive method. Predictions of spring, summer, and annual totals based on this model with ONI included are substantially more accurate than those based on the naive method; the model-based method reduces prediction error by 10–17% in the case of the spring totals, 36–40% in the case of the summer totals, and 39–49% in the case of the annual totals. The corresponding 1- and 3-month-ahead predictions, however, are only slightly more accurate than the naive predictions. Additionally, when we exclude ONI from the model, the two methods perform similarly. Plots of the observed and predicted rainfall values are available in the online supplementary material.

**Table 5.** Ratio of the prediction error (PE) based on the regression model with AR(1) errors (where ONI may be assumed known or unknown) to that based on the naive method (sample means/medians by month).

| Station | Quantity | Scale | Mean or Median | Error Measure | PE Ratio (ONI Known) | PE Ratio (ONI Unknown) |
|---------|----------|-------|----------------|---------------|----------|----------|
| Liberia | Spring total | $Y_{it}^{0.25}$ | Mean | MSE | 0.732 | 1.000 |
|         | Summer total | $Y_{it}^{0.25}$ | Mean | MSE | 0.604 | 1.000 |
|         | Annual total | $Y_{it}^{0.25}$ | Mean | MSE | 0.606 | 0.988 |
|         | 1-month-ahead | $Y_{it}$ | Median | MAE | 0.927 | 0.962 |
|         | 3-month-ahead | $Y_{it}$ | Median | MAE | 0.947 | 0.998 |
| Nicoya  | Spring total | $Y_{it}^{0.25}$ | Mean | MSE | 0.895 | 1.002 |
|         | Summer total | $Y_{it}^{0.25}$ | Mean | MSE | 0.642 | 1.000 |
|         | Annual total | $Y_{it}^{0.25}$ | Mean | MSE | 0.512 | 0.951 |
|         | 1-month-ahead | $Y_{it}$ | Median | MAE | 0.926 | 0.949 |
|         | 3-month-ahead | $Y_{it}$ | Median | MAE | 0.975 | 1.006 |

### 4.3. Tweedie Model

For predictions based on the Tweedie model, we first estimate the parameters of the model described in Section 3.3, modified to exclude the time trend—and the effects of ONI, if desired. To predict $Y_{new}$, we use the best linear predictor (e.g., [27]):

$$\hat{Y}_{new} = \mathrm{E}[Y_{new}] + \mathrm{Cov}[Y_{new}, \mathbf{Y}]' \, \mathrm{Var}[\mathbf{Y}]^{-1}(\mathbf{y} - \mathrm{E}[\mathbf{Y}]),$$

evaluated by replacing the unknown parameters with their estimates. We use $\hat{Y}_{new}$ as a point predictor of $\mathrm{E}[Y_{new} \mid \mathbf{Y} = \mathbf{y}]$.

Table 6 lists the ratios of the prediction errors based on the Tweedie model (where future ONI values may be assumed known or unknown) to those based on sample monthly means. When ONI is included, this model reduces the prediction error of annual total predictions by approximately 66%! It also reduces the prediction error of the spring total predictions by 10–27% and the prediction error of the summer total predictions by 30–39%. However, the 1- and 3-month ahead predictions are only slightly more accurate than the naive predictions. Additionally, when we exclude ONI from the model, the performances of the two methods are comparable. Plots of the observed and predicted rainfall values are available in the online supplementary material.

**Table 6.** Ratio of the prediction error (PE) based on the Tweedie model (where ONI may be assumed known or unknown) to that based on the naive method (sample means/medians by month).

| Station | Quantity | Scale | Mean or Median | Error Measure | PE Ratio (ONI Known) | PE Ratio (ONI Unknown) |
|---------|----------|-------|----------------|---------------|----------|----------|
| Liberia | Spring Total | $Y_{it}$ | Mean | MSE | 0.728 | 1.000 |
|         | Summer Total | $Y_{it}$ | Mean | MSE | 0.703 | 1.000 |
|         | Annual Total | $Y_{it}$ | Mean | MSE | 0.338 | 1.000 |
|         | 1-Month-Ahead | $Y_{it}$ | Mean | MSE | 0.931 | 0.940 |
|         | 3-Month-Ahead | $Y_{it}$ | Mean | MSE | 1.017 | 0.987 |
| Nicoya  | Spring Total | $Y_{it}$ | Mean | MSE | 0.904 | 1.000 |
|         | Summer Total | $Y_{it}$ | Mean | MSE | 0.610 | 1.000 |
|         | Annual Total | $Y_{it}$ | Mean | MSE | 0.344 | 1.000 |
|         | 1-Month-Ahead | $Y_{it}$ | Mean | MSE | 0.919 | 0.965 |
|         | 3-Month-Ahead | $Y_{it}$ | Mean | MSE | 0.963 | 0.970 |

### 4.4. Summary of Models' Predictive Performance

As expected, because of the high variability of rainfall in this region, none of the models is able to predict rainfall with high precision. That said, incorporating ONI as a predictor sometimes leads to considerable gains in prediction accuracy: all three models produce meaningfully more accurate predictions of the spring, summer, and annual totals when ONI is included.

A final point of interest is that, while the models with ONI included are able to predict the spring, summer, and annual totals with substantially greater accuracy than the naive methods, these models are able to predict 1- and 3-month-ahead rainfall with only minimally greater accuracy. The explanation seems to lie in the variation of the timing and smoothness of the spring and summer peaks. The models can predict the total rainfall during the spring and summer—and hence over the entire year—with some accuracy. However, they are not able for predicting rainfall with monthly resolution.

## 5. Discussion and Conclusions

In this paper, we analyzed the annual cycle of monthly rainfall totals for the years 1980 to 2020 at the Liberia and Nicoya stations in the Guanacaste province of Costa Rica. Our first objective was to develop a statistical model that describes this cycle and allows us to assess the long-term temporal trend and the influence of ENSO (as reflected in the ONI). Our second objective was to provide statistical methods for forecasting rainfall that account for the relatively complex annual rainfall pattern in this tropical wet–dry climatic zone. These tasks are difficult due to the many months with zero rainfall, autocorrelation, and a high degree of inter-annual variability.

To achieve our objectives, we proposed a novel model, the double-Gaussian model, as an intuitive way of describing the observed rainfall patterns. This model explicitly describes the robust, two-peak, annual rainfall pattern, leading to useful interpretations of the ONI effect. Our analysis provides no evidence of a time trend but very strong evidence of an association between ONI and spring and summer peaks of rainfall.

The double-Gaussian model relies on the assumption of normality (which is clearly violated in our setting) and independence (which may be mildly violated in our setting). We therefore checked the sensitivity of our conclusions to model misspecification by considering two alternative models. The regression model with AR(1) error relaxes the assumption of independence, and the Tweedie model relaxes the assumptions of both independence and normality. The consistency of our conclusions across models suggests that the impact of the violations of the double-Gaussian model is minimal. Therefore, we feel comfortable using it as a descriptive tool in practice.

We also used our models to predict rainfall at the two stations (training the models with data from previous months). We showed that if ONI can be reliably forecast and included as a predictor variable, predictions of spring, summer, and annual totals can be considerably more accurate than those based on sample monthly means or medians. Such predictions will be of great utility to local water managers as they plan for short-term water-management strategies [19].

We assume that the inclusion of other covariates (for example, as noted earlier, PDO and AMO indices and North Atlantic SST) would improve predictive performance. We encourage such extensions of the double-Gaussian model via structures similar to those used to incorporate ENSO forcing. In addition, lagged values of ONI could be considered, though our preliminary investigation of this issue suggests that rainfall is more strongly correlated with contemporaneous ONI values than with lagged ONI values.

Although our work was limited to point predictions of rainfall, prediction intervals are a worthwhile topic for future research. One approach would be to develop a fully parametric model based on justified assumptions. In our case, the Tweedie model, although based on reasonable assumptions, is semi-parametric and can therefore produce only point predictions, not prediction intervals. In contrast, the double-Gaussian model and regression model with AR(1) errors are parametric but are based on the flawed assumption of normality. That said, the normality assumption is reasonable if we fit those models only to rainfall from the wet months. Prediction intervals for rainfall in those months would then presumably be valid.

In the *Futuragua* project, an earlier version of the double-Gaussian model [19], with its high degree of interpretability, proved to be well-suited to developing a set of rainfall scenarios ranging from extreme La Niña to extreme El Niño conditions. These rainfall

scenarios were used in turn to drive a hydrological watershed model in Guanacaste and allowed projection of surface water and groundwater supplies under different ENSO conditions [28]. The results were used to inform sustainable water management in the region and support local decision makers in preparing, for instance, for drier El Niño conditions. The extension of the model developed in the present paper will presumably allow refinements of these management strategies. Furthermore, we have shown that when ONI projections (which are typically available 3–6 months in advance) are incorporated, our models can provide substantially more accurate forecasts of spring, summer, and annual totals than naive methods. These forecasts can provide valuable planning information to local water managers.

**Supplementary Materials:** The following supporting information can be downloaded at: https://www.mdpi.com/article/10.3390/w15040700/s1, Supporting Information for "Statistical Modelling of the Annual Rainfall Pattern in Guanacaste, Costa Rica".

**Author Contributions:** Conceptualization, R.M.A., O.H. and D.S; methodology, R.M.A., O.H., N.M. and D.S.; software, R.M.A., O.H. and N.M.; formal analysis, R.M.A., O.H., N.M. and D.S.; data curation, N.M.; writing–original draft preparation, R.M.A., O.H. and D.S.; writing—review and editing, R.M.A., O.H., N.M. and D.S.; funding acquisition, R.M.A. and D.S.; project administration, D.S. All authors have read and agreed to the published version of the manuscript.

**Funding:** D.S. secured funding for the *FuturAgua* project from NSERC through an International Opportunities Fund within the G8 Research Councils Initiative on Multilateral Research Funding and Belmont Forum Grant opportunity. R.A. was funded by NSERC grant number RGPIN-2018-04304. O.H. was funded by CANSSI. N.M. was funded by the *FuturAgua* grant.

**Data Availability Statement:** Code and data will be made available on GitHub.

**Acknowledgments:** We are extremely grateful to IMN for giving us access to station data. This access was facilitated by an important *Futuragua* collaborator, *Área de Conservación Tempisque*. We also thank William Welch of the University of British Columbia for his helpful input to this project.

**Conflicts of Interest:** The authors declare no conflict of interest.

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
