# Peer review of "Statistical Modelling of the Annual Rainfall Pattern in Guanacaste, Costa Rica"

_water, doi:10.3390/w15040700_

Round 1

Reviewer 1 Report

1) The abstract needs to be reorganized. The three statistical models need to be specifically pointed out.

2) In “4. Rainfall Forecasting” part, it forecasted the spring and summer rainfall peaks over the years 2006-2020 in Liberia and 2007-2016 in Nicoya. However, the data of these years were included in the model development. It cannot explain the predictability of such methods. It should be fully explained in the discussion.

3) In P4 lines 174-175, “Choosing these two stations affords an opportunity to detect any effects of local topographic rainfall forcing embedded in larger-scale effects". Please give an answer to this questions by comparing the parameter results of the two sites. 

 4) In 3. Statistical Models and Estimation of the ONI Effect and Long-Term Time Trend part, please add a part to point out the differences and similarities in the results of the three methods, as well as the the advantages and disadvantages of each method.

Reviewer 2 Report

Summary:

Authors discussed three statistic models for annual rainfall pattern and prediction two meteorological observations, Nicoya station and Liberia station. The paper is well written. I would suggest authors to modify the paper according to my minor comments to make the manuscript more complete and solid. The paper could be publishable in Water with minor revisions.

Minor comments:

1.      My comments are for the section 4 (rainfall forecasting): [Table 4-6]

When authors list the ratio of prediction error (PR) of (1) spring total, (2) summer total, (3) annual total, (4) 1-month-ahead, (5) 3-month-ahead. I suggest authors list the exact values of (1) mean/median of observation rainfall and prediction rainfall and (2) error measure: MSE/MAE and bias. The bias has positive or negative value which can reveal the prediction rainfall is higher or lower than observation rainfall.

Reviewer 3 Report

This manuscript presents a study on statistical modeling of the annual rainfall pattern in Guanacaste, Costa Rica. The model provides an intuitive way of describing the seasonality of rainfall, the inter-annual variability of the cycle, and variability due to the monthly oceanic Nino index. These forecasts could provide valuable planning information to local water managers. Some comments on the present manuscript are as follows.

 1. The manuscript has few figures (with only two figures). The results predicted by the models may be shown by some new figures.

 2. Line 52: SST anomalies in the Eastern Pacific and tropical North Atlantic Oceans have an effect on rainfall. The tropical North Atlantic SST anomalies may have important climate effects (Wu and Zhang, 2010; Wang et al., 2017; Zhao et al., 2022). It is suggested that the tropical North Atlantic SST anomalies may be included in the prediction model. At least, some discussions on this point should be added in the manuscript.

 Wu R. and L. Zhang, 2010: Biennial relationship of rainfall variability between central America and Equatorial South America. Geophysical Research Letters, 37: L08701.

 Wang L., J.Y. Yu, H. Paek,2017: Enhanced biennial variability in the Pacific due to Atlantic capacitor effect. Nature Communications ,8 ,14887.

 3. The PDO and AMO index may also be included in the model to capture the interdecadal time scale signals, like in other studies (e.g., Zhao et al., 2022).

 Zhao J., Zhan R., Wang Y., Jiang L., Huang X. 2022: A multiscale-model-based near-term prediction of tropical cyclone genesis frequency in the Northern Hemisphere. Journal of Geophysical Research: Atmosphere. 127, e2022JD037267.  

https://doi.org/10.1029/2022JD037267.

Round 2

Reviewer 3 Report

The authors have revised the manuscript according to previous comments. There are no other comments.